# Using MaxEnt Model to Predict the Potential Distribution of Three Potentially Invasive Scarab Beetles in China

**DOI:** 10.3390/insects14030239

**Published:** 2023-02-27

**Authors:** Shuangyi Wang, Yuanyuan Lu, Mengyang Han, Lulu Li, Ping He, Aimin Shi, Ming Bai

**Affiliations:** 1College of Life Science, China West Normal University, Nanchong 637002, China; 2Key Laboratory of Zoological Systematics and Evolution, Institute of Zoology, Chinese Academy of Sciences, Beijing 100101, China; 3School of Agriculture, Ningxia University, Yinchuan 750021, China; 4Faculty of Geographical Science, Beijing Normal University, Beijing 100875, China; 5College of Plant Protection, Hebei Agricultural University, Baoding 071001, China; 6Northeast Asia Biodiversity Research Center, Northeast Forestry University, Harbin 150040, China; 7College of Life Sciences, University of Chinese Academy of Sciences, Beijing 100049, China

**Keywords:** quarantine, invasive insect, potential distribution, MaxEnt model

## Abstract

**Simple Summary:**

Based on the occurrence records and bioclimatic variables, the MaxEnt model was used to predict the potential geographical distribution of *Heteronychus arator*, *Oryctes boas* and *Amphimallon majale* worldwide, and we also discuss the possible distribution of *Popillia japonica*, *Heteronychus arator*, *Oryctes monoceros*, *Oryctes boas* and *Amphimallon majale*. The results showed that every continent has potential distribution areas for these species. In China, the potential distribution of *Popillia japonica* and *Amphimallon majale* was mainly concentrated in central eastern regions, while the presence of *Heteronychus arator* and *Oryctes boas* was mainly concentrated in the southwest areas. *Oryctes monoceros* has no possible distribution at present. Based on the predicted results, we recommend that the government carry out species monitoring to minimize financial losses.

**Abstract:**

A hot topic in recent years is the prediction of the potential distribution of possible invasive insects. China is facing a great challenge due to invasive insects. Scarab beetles are a highly diverse group, and many of them are well-known invasive insects. Here, in order to prevent the invasion of scarab beetles in China, we screened the invasive insects globally and obtained a preliminary database of quarantine or invasive scarab beetles. From the obtained database, we selected the top five species (*Popillia japonica*, *Heteronychus arator*, *Oryctes monoceros*, *Oryctes boas* and *Amphimallon majale*) to discuss and analyzed the potential distribution of three species that have not invaded China by using the MaxEnt model. The prediction results show that every continent has potential distribution areas for these species. Specifically within China, *Popillia japonica* and *Amphimallon majale* were mainly concentrated in east central regions and *Heteronychus arator* and *Oryctes boas* were mainly distributed in the southwest areas, while *Oryctes monoceros* has no suitable area. Notably, Yunnan, Hunan, Jiangxi and Zhejiang province had a high risk of invasion. In general, local agriculture, forestry and customs departments in China should pay more attention to monitoring for the prevention of infestation by invasive insects.

## 1. Introduction

Invasive alien species threaten biodiversity around the world. Increasing global trade and environmental changes have facilitated the arrival and establishment of invasive species, and as a result, there is a global push toward research on controlling [1]. Various invasive alien species have been introduced to new areas by the trade and transportation of goods, and they have thrived outside of their native regions [2]. China is also facing a great challenge due to biological invasion, especially from insects [3]. For example, the General Administration of Customs of the People’s Republic of China (http://dzs.customs.gov.cn/ (accessed on 26 September 2022)) issued a list of 446 species titled the “Catalogue of quarantine pests for import plants to the People’s Republic of China”, including 148 species of insects, accounting for approximately 33% of the total [3]. In addition, the majority of the species quarantined at Chinese ports are from cargo quarantines, with insects being the most common [4]. To reduce the losses caused by invasive alien insects, it is generally believed that preventing an outbreak is more feasible and economical than controlling them [5,6].

Scarab beetles (Coleoptera: Scarabaeoidea) are a highly diverse group of insects with a number of feeding habits and a wide distribution [7]. Furthermore, many phytophagous scarab beetles are well-known invasive alien insects, and, whether as adults and larvae, or both, they attack crop plants, nurseries and forests. For example, *Popillia japonica* is a polyphagous, widespread and destructive pest that has over 300 species of hosts [8]. According to statistics, it has spread to many countries and regions in North America and Europe, and over USD 460 million is spent annually toward its control [9,10,11,12]. In addition, *Oryctes monoceros*, *Holotrichia serrata*, *Phyllophaga smithi* and *Adoretus versutus* are also important quarantine scarab beetles. However, the current knowledge about beetles is insufficient, and only a few scarab beetles have received attention [12,13]. Herein, we will focus on the distribution of scarab beetles that may pose serious hazards: *Popillia japonica* (*P. japonica*), *Heteronychus arator* (*H*. *arator*), *Oryctes monoceros* (*O*. *monoceros*), *Oryctes boas* (*O. boas*) and *Amphimallon majale* (*A. majale*). Exploring the potential distribution of these insects worldwide, especially in China, the generated information would significantly improve understanding of the possible invasion risk [14].

By understanding the basic biological information of these five species, we can obtain a better understanding of their hazards and the significance of predicting potential suitable areas. The first species is *P. japonica*, which is a highly polyphagous invasive scarab [12]. It has over 300 reported host plants [10,12], including turfgrass, landscapes, nursery crops and Acer spp. [9]. Both adults and larvae are pests. Adults damage the foliage, flowers and fruit surfaces of many valuable plant species, and the larvae damage the roots of various plants and grasses, which often leads to the destruction of turfgrass [9,11]. The second species is *H*. *arator*, which is an invasive and polyphagous pest of quarantine importance [15]. *H. arator* attacks economically important food crops, such as maize, barley, sugarcane, potatoes, grapes and forage grass [15,16]. Adults and larvae feed on organic matter and plant roots—adults always nibble on plant stems and roots slightly below the soil surface, while larvae feed on detritus and plant roots below the ground [17,18]. The third species is *O*. *monoceros*, which is a serious pest of palms such as *Phoenix dactylifera* and *Cocos nucifera* [13,19]; it also attacks non-palm hosts such as *Saccharum officinarum* and *Musa paradisiaca* [20]. Both adults and larvae are pests. While the larvae develop in decomposing organic matter, adults feed inside the unopened fronds and meristems of palms [13,19,21]. The fourth species is *O. boas,* which is mainly harmful to economic crops, such as coconut, palm and date [19,22], and both adults and larvae are pests. The larvae pose serious damage but the adults do not, and the larvae are more commonly found in rubbish heaps composed of decomposing vegetable matter and manure [19,22]. The last species is *A. majale* and it is mainly harmful to meadows, pastures, winter grains, ornamental nurseries and home lawns [23]. In recent years, corn has been documented as a potential or minor host [24], and the adults do not injure plants but the larvae are the most common and destructive white grub species [25].

In order to effectively prevent the invasion and spread of potentially invasive scarab beetles in China, we need to screen the main invasive insects globally. Therefore, we collected a list of quarantine or invasive insects in various countries and regions around the world. For instance, a report on the status of biological invasions and their management in South Africa in 2019 was published by the South African National Biodiversity Institute, which included a list of alien species with 429 insects (1 scarab beetle) that provided the basis for further pest control, and this was considered a step toward a national registry of alien species (https://www.gov.za/ (accessed on 15 December 2021)). After the collection, we obtained a preliminary database of quarantine or invasive scarab beetles from 138 countries and regions, which was not available previously. Based on the database, species were selected for analysis, and their potential distribution areas were discussed.

Species distribution models are widely used inferential tools that use geographic data and climatic variables to estimate the environmental requirements that predict a species’ potential distribution [26]. Commonly used species distribution models are the Match Climates Regional Algorithm (CLIMEX), Genetic Algorithm for the Rule Set Production (GARP), and Maximum Entropy (MaxEnt) model. Each model has a different theoretical basis, data requirements and analysis methods [14]. Among these models, the MaxEnt model performs better even in the absence of species occurrence records, and it differs from other models in that it operates quickly and simply, so it has become an ideal prediction tool to analyze the distribution or potential distribution [14,27,28]. In recent years, the MaxEnt model has been used to model the distribution of the protection of economic species [29], prevention of invasive species [30], and endangered and threatened species [31]. Thus, in order to study the potential distribution of the selected species, we used the MaxEnt model for analysis.

In this study, the top five species of the database with no records in China were discussed and analyzed. Based on the prediction of the potential distribution here, potential optimal/moderate/marginal suitability and unsuitable areas of the unstudied *H*. *arator*, *O. boas* and *A*. *majale* could be determined. Furthermore, the results obtained in this study will help environmental protection and quarantine agencies to implement quarantine measures that will reduce the possibility of invasion of these insects in China and other countries. In addition, the database introduced in this paper can better provide data and information support systems for ensuring national biosecurity.

## 2. Materials and Methods

### 2.1. Data Source

We obtained a list by searching the Food and Agriculture Organization of the United Nations (https://www.ippc.int/zh/ (accessed on 2 April 2021)), the Global Invasive Species Database (http://www.iucngisd.org/gisd/ (accessed on 1 June 2022)), and the official websites of the agriculture, forestry and other relevant departments of the priority countries and regions one by one, and focusing on the major economies and the countries and regions that cooperate with the Chinese Belt and Road Initiative. After screening, we obtained a list of quarantine or invasive scarab beetles from 138 countries and regions. Then, according to the frequency of each species being classified as quarantine or invasive alien insects by several countries and regions, we sorted the list and collected the host, hazard and other information, so as to obtain a preliminary database of quarantine or invasive scarab beetles. The top five pests in the database that had not invaded in China were *P. japonica* (61), *H*. *arator* (13), *O*. *monoceros* (12), *O. boas* (11) and *A. majale* (7). Since *P*. *japonica* and *O*. *monoceros* have already been studied [12,13], this study will focus on *H*. *arator*, *O. boas* and *A. majale*
Appendix A.

### 2.2. Occurrence Data

Occurrence records of *H*. *arator*, *O. boas* and *A*. *majale* were obtained from the GBIF database (Global Biodiversity Information Facility, https://www.gbif.org/ (accessed on 12 October 2022)), the national biosafety basic data information resource platform (http://www.pestchina.com/#/ (accessed on 12 October 2022)), ISC database (Invasive Species Compendium, https://www.cabi.org/isc (accessed on 12 October 2022)), EPPO database (European and Mediterranean Plant Protection Organization, https://gd.eppo.int/ (accessed on 12 October 2022)) and the related literature [17,26,32,33,34,35]. Records lacking geographic coordinates were georeferenced in Google Maps (http://www.google.cn/intl/ zh-CN/earth/ (accessed on 15 October 2022)) and Geonames (http://www.geonames.org/ (accessed on 15 October 2022)) [36]. To avoid over-fitting due to sampling deviation, we subsampled points at a 2.5 arc grid to reduce sampling bias and to minimize the possible effects of spatial autocorrelation [37]. Finally, 351 records for *H*. *arator*, 177 records for *O. boas* and 463 records for *A*. *majale* were used to build the predictive model (Figure 1 and Appendix A).

### 2.3. Bioclimatic Variables

The survival of invasive insects is closely related to bioclimatic factors. Bioclimatic variables are widely used to predict the distribution of invasive alien insects at regional and global scales [14]. In total, 19 WorldClim bioclimatic variables, with averages for the years 1950–2000, with a spatial resolution of 2.5 arc min, used for modeling, were obtained from the WorldClim Database (Version 2.1, http://www.worldclim.org (accessed on 12 October 2022)) [38]. Multicollinearity among the bioclimatic variables may have a negative impact on the relationship among species distribution [39]. To establish a high-performance model with fewer variables, the Pearson correlation coefficients of the cross-correlations among the 19 bioclimatic variables were calculated using the SPSS software (Version 20.0), and the Pearson correlation coefficient (threshold) was included in parentheses [40]. Thresholds of *O. boas* were chosen differently from those of *H*. *arator* and *A*. *majale* because of the high correlation between the bioclimatic factors. Only one variable from each set of the highly cross-correlated variables was retained for further study; then, we selected seven variables (Pearson > 0.8) to be included for the *H*. *arator*, five variables (Pearson > 0.95) for *O. boas* and eight variables (Pearson > 0.8) for *A*. *majale* for analysis [41] (Table 1 and Appendix A).

### 2.4. Modeling Methods

The map used in this study was from the Resource and Environment Science and Data Center (https://www.resdc.cn/ (accessed on 8 November 2021)). We used the MaxEnt model (version 3.4.1, http://biodiversityinformatics.amnh.org/open_source/maxent/ (accessed on 8 November 2021)) to predict the potential distribution of *H*. *arator*, *O. boas* and *A*. *majale* globally and in China. For the study of China, the model is a local amplification of the world results. The selected geographical distribution records and bioclimatic variables were imported into the MaxEnt model for analysis. The parameters of the MaxEnt model were set as follows: the available features were autofeatures including linear, quadratic, product, threshold and hinge [42], the MaxEnt model was run with 25% randomly selected data for testing, and the remaining 75% of data were used for training [41,43]. The logistic outputs were used in the MaxEnt model, which generated a continuous map with an estimated probability of presence between 0 and 1 [44]. The rest of the parameters were set by default. This analysis provided the average contribution rate of each variable to the model, the jackknife analysis of the contribution of each variable to the model and the response curve with a standard deviation error bar [42]. Then, we input the results into the ArcGIS software (Version 10.7) for visual expression and used the Natural Breaks [Jenks] method to reclassify them into four different levels: optimal suitability areas, moderate suitability areas, marginal suitability areas, and unsuitable areas [43,45].

In addition, the significance of the variables contributing to the three species’ distribution was represented by the average values of the area under the receiver operating characteristic (ROC) curve (AUC) of 10 model iterations [46]. The ROC curve is a comprehensive indicator reflecting the continuous variables of the sensitivity and specificity of the model, and the AUC value (0–1) is taken as a measure of the accuracy of the model [45]. Generally, AUC < 0.5 means that the results are not credible, 0.5 ≤ AUC < 0.7 denotes poor performance, 0.7 ≤ AUC < 0.9 denotes average results, and 0.9 ≤ AUC < 1 denotes high performance [42].

## 3. Results

### 3.1. Modeling Results Validation

The AUC value in the MaxEnt model was used to evaluate model performance; the larger the AUC value, the greater the correlation between bioclimatic variables and the predicted distribution area, and the better the model performs [47]. The average AUC values for 10 repetitions for *H. arator*, *O. boas* and *A. majale* were 0.977, 0.959 and 0.972, respectively, indicating that the model yielded highly reliable results in predicting the potential distribution [40] (Appendix A). 

### 3.2. Key Bioclimatic Variables

According to the relative contribution of each bioclimatic variable in predicting the potential distribution of *H*. *arator*, *O. boas* and *A*. *majale*, we obtained some available information: the most important environmental variable affecting the potential distribution of *H. arator* and *O. boas* was Bio3 (47.9% and 67.3%), with the highest contribution; and the most crucial environmental variable of *A. majale* was Bio17 (52%) (Table 2). The training results showed that Bio1 had the highest gain on *H*. *arator* when used alone, and it also appeared to have the maximum gain reduction when it was omitted (Figure 2). For *O. boas,* Bio3 had the greatest effect when used alone, and it also had the greatest reduction in gain when it was omitted. Meanwhile, Bio17 affected *A*. *majale* with the maximum gain when used in isolation, and Bio2 decreased the gain the most when it was omitted. Overall, the training results were consistent with the test data. They showed that Bio1 and Bio3 were the most important bioclimatic variables in explaining the potential distribution of *H*. *arator*; for *O. boas*, it was Bio3, and for *A*. *majale*, the most important were Bio17 and Bio2. The response curve between bioclimatic variables and the probability of a species’ presence reflects the relationship [48]. As shown in Appendix A, with the temperature and precipitation change, the probability of the occurrence of a species was constantly changing, and the relatively suitable environmental conditions also corresponded to the potential distribution in China. The optimum values and range are as follows. For *H*. *arator*, the optimal value of Bio1 was 15.0 °C, and the suitable range was 2.11~28.5 °C; the optimal value of Bio3 was 47.7 mm with the suitable range of 31.0~100.1 mm. For *O. boas*, the optimal value of Bio3 was 56.2 mm, and the suitable range was 35.7~100.1 mm. For *A*. *majale* the optimal value of Bio2 was 10.2 °C, and the suitable range was 1.33~16.4 °C; the optimal value of Bio17 was 145.2 mm with the suitable range of 0~695.2 mm.

### 3.3. Potential Geographical Distribution

The global prediction results of *H*. *arator*, *O. boas* and *A*. *majale* were fit to the known distribution of them in both their native and invaded ranges, which reflected the detection efficiency of our models. *O*. *boas* had the largest potential distribution area among the three species (Table 3), accounting for 14.8% of the world land area, which was successfully identified by the model of world distribution, with areas of optimal suitability mainly in southeast Asia, central and southern Africa, the southeast coast of South America, Australia and New Zealand. Meanwhile, *H*. *arator* had a relatively narrower distribution range than *O*. *boas,* accounting for 10.5%. In particular, there were clear differences in parts of Africa, Asia and Oceania. Moreover, *A. majale* covers the smallest areas of them, accounting for 7.9%. The optimal suitability of *A. majale* was mainly in North America and Western Europe (Figure 1 and Figure 3A–C). The percentage distribution area worldwide and the optimal, moderate and marginal suitability areas are presented in Table 3.

The potential distribution of *H*. *arator* and *O. boas* in China was mainly concentrated in the southwest areas and they had overlapping areas in some provinces, and *A*. *majale* was relatively widespread in the central and eastern regions (Figure 3D–F). Among them, *A*. *majale* covers the largest potential distribution area of approximately 9.1 × 10^5^ km^2^, accounting for 9.5% of the Chinese land surface area, and the optimal and moderate suitability areas included Hunan, Zhejiang, Jiangxi, Hubei, Guizhou, Guangxi, Guangdong, Fujian, Anhui, Jiangsu, Shanghai. The *O. boas* covers about 5.3 × 10^5^ km^2^, account for 5.6%, and the optimal and moderate suitability areas involved Yunnan, Sichuan, Xizang and Hainan. Moreover, *H*. *arator* covers the smallest possible distribution area of approximately 3.6 × 10^5^ km^2^, accounting for 3.8%, and the optimal and moderate suitability areas were mainly in Yunnan and Sichuan (Figure 3D–F and Table 3). 

## 4. Discussion

The preliminary list of quarantine or invasive scarab beetles in this study is the first organized database that includes information on their occurrence records, hosts, and damage. This database is also unique and comprehensive in that it combines almost all available lists of quarantine or invasive insects from 138 countries and areas globally. Based on the database, we analyzed and discussed the top five species that have not invaded China, which are *P. japonica*, *H*. *arator*, *O*. *monoceros*, *O. boas* and *A. majale*. By using the MaxEnt model and ArcGIS software, we obtained the potential distribution of *H. arator*, *O. boas* and *A. majale* in the world. The AUC values for all of them were above 0.9, which indicates that the ManEnt model is useful here. Combining the prediction results from this and the previous studies of Kistner-Thomas (2019) and Aidoo (2022) for *P. japonica* and *O. monoceros,* the potential suitable areas of these five species were obtained. In addition, considering the destructive impacts in native or invasive areas of these insects, their spread to other parts of the world should be prevented as early as possible.

The model results showed that the potential distributions of these five species were relatively wide (Figure 3A–C), with each continent at risk of invasion by one or more species (including the origin locations), and more attention should be paid to these species, especially in the countries and regions that frequently trade with China [49,50]. Within China, the potential distribution areas of these species were mainly in the southwest and eastern central areas. These regions have high biodiversity, complex environments, relatively high temperatures and more rainfall, making them more suitable for the survival of insects, including alien insects [49]. *P*. *japonica* and *A*. *majale* were mainly concentrated in the eastern central regions, and *P*. *japonica* was more widespread covering both hilly and plain areas. *H.arator* and *O. boas* were distributed in the southwest areas and had a similar distribution; between them, *O. boas* had a wider potential distribution, mainly in Yunnan. Meanwhile, *O*. *monoceros* had no potential range in China, but Aidoo (2022) found that extensive areas of Asia were highly favorable to the pest, so it may reach Yunnan from the southern coast of Asia through customs ports and then spread in China; thus, it should still be given some level of attention. In all possible areas, the medium- and high-risk provinces, such as Yunnan, Hunan and Guizhou, should strengthen their quarantine requirements and should focus on enhanced monitoring by customs, forestry and other departments. For instance, Yunnan might focus on *P*. *japonica*, *H*. *arator* and *O. boas*, as the hot and humid environment in Yunnan province is ideal for their survival.

In addition, we found that temperature and precipitation are the two main environmental factors affecting the potential distribution of these five species Appendix A [12,13]. Both in the world and in China, they show similar characteristics to their native environments, including relatively high temperatures and humidity and low temperature extremes. The potential distribution of these five species is relatively large (Figure 1 and Figure 3D–F); the chances of their survival and establishment are very high in some areas. Therefore, they also have the possibility of invading China and can be considered potential invasive insects. Furthermore, the range of species will probably expand and shift due to the impact of global warming [51], and it is possible that the moderate and marginal areas may also become optimal suitability areas, and unsuitable areas may also become suitable areas in the future. This means that China will face a more severe situation involving the invasion of alien harmful insects. Thus, it is necessary to screen species with an invasion risk continuously and to regularly check the most suitable areas. Moreover, the distribution of species can also be affected by the host distribution, geographical conditions, natural enemies, and human activities, which should be considered in further research [52,53]. For instance, *H*. *arator* mainly damages Gramineae plants and economic crops, which are widely cultivated in China Appendix A. This further validates the importance of prevention.

A large number of insects are quarantined at customs ports every year. Thus, in addition to these five species discussed and analyzed in this study, there are more species that may also carry potential invasion risks for China. If we rely solely on customs quarantine for the control of invasive alien insects when conducting research and prevention, we will only occupy a passive position and delay the adoption of preventive measures. However, when we build and continuously complete the database of quarantine or invasive scarab beetles and use this information in preventive measures, we will be able to constantly convert the passive information into active action. In the future, further data supplements will focus on the new, harmful insects intercepted at customs ports and detected by the agriculture and forestry departments of each province or the sudden outbreak of harmful scarab beetles abroad with the possibility of global spread. At present, a serious problem in the field of science and technology development in China is the imperfect construction of data and information systems. Compared with developed countries, there is still a large gap. In addition, database construction is a fundamental task that needs long-term investment; if we can build and improve the database of quarantine or invasive scarab beetles successfully, this will become an important step in the process of ecological protection and sustainable development in China and around the world [54]. Among the existing databases, the “Database of invasive alien species in China” and the “National Pest Quarantine Information System in China” are relatively complete and functional databases, with implications for pest management around the world [54,55]. However, we believe that the database of quarantine or invasive scarab beetles that we have built here and the work that we will conduct with the database may be a powerful tool enabling us to take one more steps toward preventive measures.

## Figures and Tables

**Figure 1 insects-14-00239-f001:**
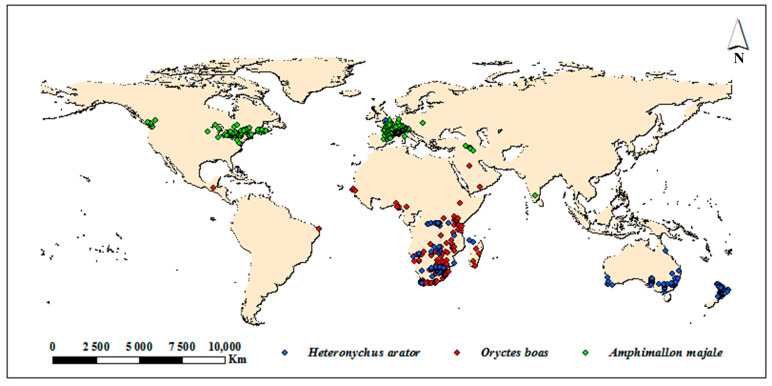
Occurrence records of *Heteronychus arator*, *Oryctes boas* and *Amphimallon majale*. Three different-colored dots indicate distribution records.

**Figure 2 insects-14-00239-f002:**
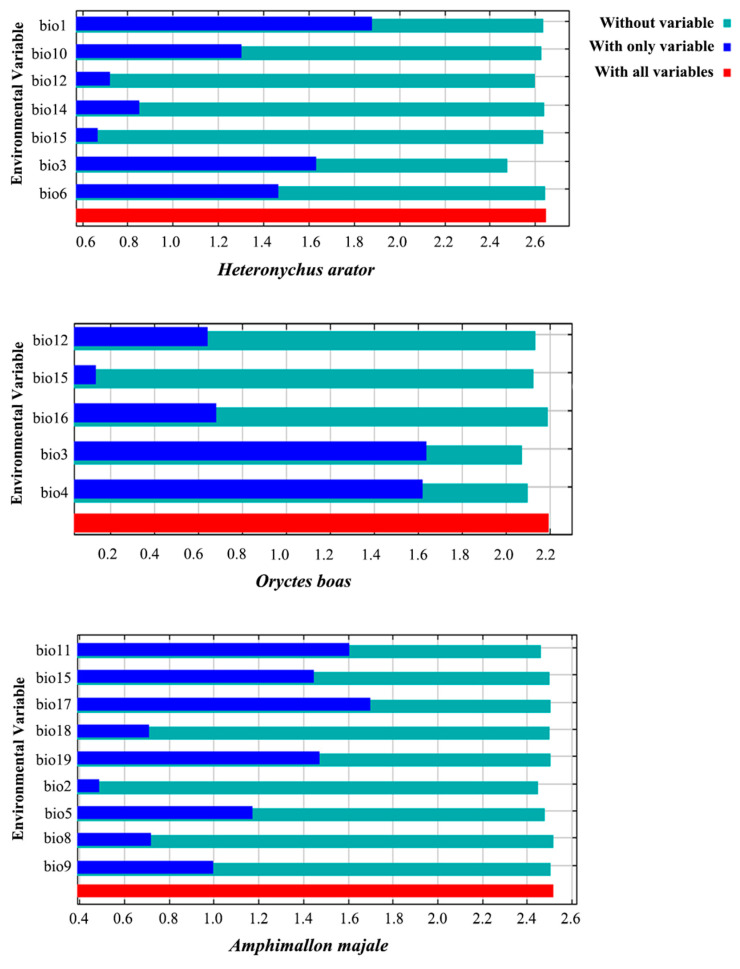
Jackknife test results of bioclimatic variables importance for *Heteronychus arator*, *Oryctes boas* and *Amphimallon majale*. Green, blue and red bars represent running the MaxEnt model without the variable alone, with only variable, and with all variables, respectively.

**Figure 3 insects-14-00239-f003:**
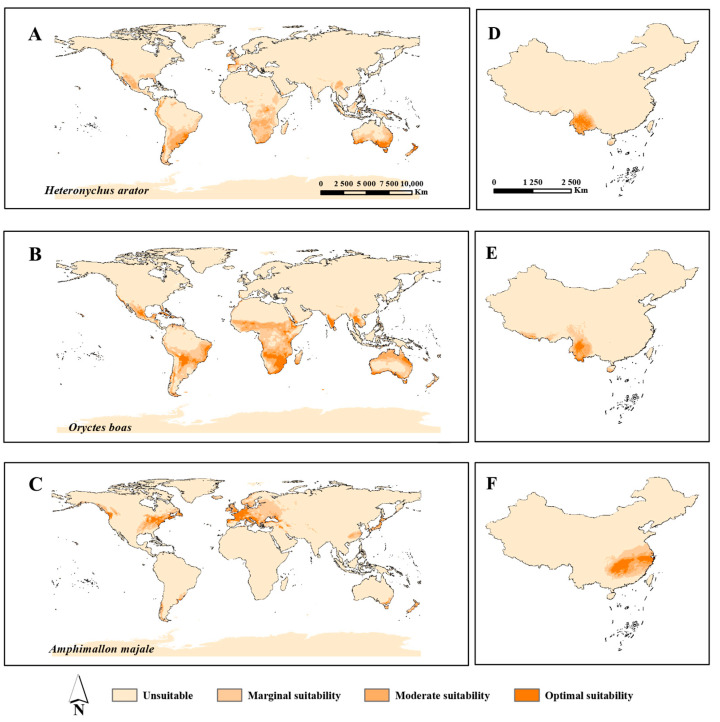
Potential distribution of *Heteronychus arator*, *Oryctes boas* and *Amphimallon majale* in the world (**A**–**C**) and in China (**D**–**F**). (**A**,**D**): For *Heteronychus arator*, the range of the suitablility (occurrence probability) from unsuitable to optimal suitability is 0~0.05, 0.05~0.18, 0.18~0.40, and 0.40~0.82 in the world and 0~0.04, 0.04~0.14, 0.14~0.25, and 0.25~0.39 in China; (**B**,**E**): For *Oryctes boas*, from unsuitable to optimal suitability is 0~0.08, 0.08~0.26, 0.26~0.49, and 0.49~0.88 in the world and 0~0.02, 0.02~0.10, 0.10~0.19, and 0.19~0.53 in China; (**C**,**F**): For *Amphimallon majale*, from unsuitable to optimal suitability is 0~0.06, 0.06~0.21, 0.21~0.43, and 0.43~0.85 in the world and 0~0.03, 0.03~0.11, 0.11~0.20, and 0.20~0.38 in China.

**Table 1 insects-14-00239-t001:** Selected bioclimatic variables of *Heteronychus arator*, *Oryctes boas* and *Amphimallon majale* in this study.

Variable	Variable Description	*Heteronychus arator*	*Oryctes boas*	*Amphimallon majale*
Bio1	Annual mean temperature	√		
Bio2	Mean diurnal range			√
Bio3	Isothermality (Bio2/Bio7) (×100)	√	√	
Bio4	Temperature seasonality	√	√	
Bio5	Max temperature of the warmest month			√
Bio6	Min temperature of coldest month			
Bio8	Mean temperature of the wettest quarter			√
Bio9	Mean temperature of the driest quarter	√		√
Bio10	Mean temperature of warmest quarter			
Bio11	Mean temperature of the coldest quarter			√
Bio12	Annual precipitation		√	
Bio14	Precipitation of driest month	√		
Bio15	Precipitation seasonality	√	√	
Bio16	Precipitation of wettest quarter		√	
Bio17	Precipitation of driest quarter			√
Bio18	Precipitation of the warmest quarter	√		√
Bio19	Precipitation of the coldest quarter			√

**Table 2 insects-14-00239-t002:** Key bioclimatic variables and contributions of *Heteronychus arator*, *Oryctes boas* and *Amphimallon majale*.

Variable	Percent Contribution		
	*Heteronychus arator*	*Oryctes boas*	*Amphimallon majale*
Bio1	21.4		
Bio2			3.2
Bio3	47.9	67.3	
Bio4	4.5	12.7	
Bio5			8.1
Bio9	5.1		0.7
Bio11			16.6
Bio12		13.8	
Bio14	17.7		
Bio15	0.4	3.2	4
Bio16		3	
Bio17			52
Bio18	3		2.6
Bio19			12.8

**Table 3 insects-14-00239-t003:** Percentage of the potential distribution area for *Heteronychus arator*, *Oryctes boas* and *Amphimallon majale*.

Percentage	Species (World/China)		
	*Heteronychus arator*	*Oryctes boas*	*Amphimallon majale*
Potential distribution area	10.5%/3.8%	14.8%/5.6%	7.9%/9.5%
Optimal suitability area	0.8%/1.1%	2.3%/0.8%	1.2%/2.4%
Moderate suitability area	3.0%/1.4%	5.3%/1.6%	2.0%/2.9%
Margin suitability area	6.7%/1.3%	7.2%/3.2%	4.7%/4.2%

## Data Availability

The data presented in this study are available on request from the corresponding author.

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
