# Peer review of "Using MaxEnt Model to Predict the Potential Distribution of Three Potentially Invasive Scarab Beetles in China"

_insects, 2023, doi:10.3390/insects14030239_

Round 1

Reviewer 1 Report

The manuscript uses pooled information from different quarantine databases and predictive models to identify the areas at risk of invasion from scarab beetles. The manuscript is not very attractive in its current form and will need more editing and proofreading. I have highlighted some areas that need editing and rephrasing. Comments are in the attached Pdf. 

Author Response

Cover letter

Thanks to the reviewers for their important suggestions on the effective improvement of this article. The relevant content has been revised, please see the following for details.

This article has been revised by the language editor.

Suggestion 1: The manuscript uses pooled information from different quarantine databases and predictive models to identify the areas at risk of invasion from scarab beetles. The manuscript is not very attractive in its current form and will need more editing and proofreading.

Reply: Thanks for your suggestion, which is very useful to us. At present, our article has been revised for many times, and we have made further explanations for the corresponding contents of each part. And current work is a preliminary analysis and use of the database of quarantine or invasive scarab beetles. In the future, the next research and use will focus on the depth of the study to make better use of the database.

Suggestion 2: I have highlighted some areas that need editing and rephrasing. Comments are in the attached Pdf.

Reply: Thanks for your suggestion, which is very useful to us. Where the manuscript was revised by the reviewer, it has been revised accordingly one by one. And the sentences that need to be rephrase have been done.

See page 1, 2, 3, 4, 5, 7, 8, 9 for details.

Suggestion 3: those multiple → these species

Suggestion 4: Delete “but more attention should be pay”, “as early”, “also” and “Oryctes monoceros had no records now”

Reply: Thanks for your suggestion. Now it has been revised.  Page 1

Suggestion 5: Rephrase this sentence “Overall, the southwest and east-central regions of China were more suitable for invasive insects, which should be paid more attention by local agriculture, forestry and customs departments”

Reply: Thanks for your suggestion, we have made the following modifications. Change to “In general, local agriculture, forestry and customs departments in the southwest and east–central regions of China should pay more attention to the prevention of infestation by insects.”  Page 1

Suggestion 6: China also → China is also; diversity → diverse; including a variety of → with a number of; Delete “numbers of”; And → Furthermore;

Reply: Thanks for your suggestion. Now it has been revised.  Page 2

Suggestion 7: over U.S. $460 million was spent annually to repair over → USD 460 million is spent annually towards its control

However, the relevant research is insufficient, and only a few scarab beetles have received attention → However, the current knowledge about beetles is insufficient, and only a few scarab beetles have received attention

Reply: Thanks for your suggestion. Now it has been revised.  Page 2

Suggestion 8: Delete “they are”

Suggestion 9: Common used species distribution models are Match Climates Regional Algorithm (CLIMEX), Genetic Algorithm for the Rule Set Production (GARP), MaxEnt model and so on → Commonly used species distribution models are the Match Climates Regional Algorithm (CLIMEX), Genetic Algorithm for the Rule Set Production (GARP), and Maximum Entropy (MaxEnt) model. Each model has a different theoretical basis, data requirements and analysis methods

Reply: Thanks for your suggestion. Now it has been revised.  Page 2

Suggestion 10: Delete “In this study, the top five species of the database that no records in China had been discussed and analyzed.”

Reply: Thank you very much, we have made the following modifications. Change to “In this study, the top five species of the database with no records in China were discussed and analyzed.”We think this sentence can better lead to the content to be studied below, so we want to retain it.  Page 2

Suggestion 11: Accordingly, combined with the possible distribution of P. japonica and O. monoceros that has been analyzed

Reply: Thank you very much, we have made the following modifications. Change to “Further, the obtained results were combined and compared with those for P. japonica and O. monoceros, which have been analyzed previously.”  Page 2

Suggestion 12: preventive measures can be formulated and implemented as soon as possible to reduce economic losses through prevention and ensure the sustainability of Chinese ecological environment.

Reply: Thank you very much, it is very helpful. Change to “and this will help environmental protection and quarantine agencies to implement quarantine measures that will reduce the possibility of invasion from these insects in China and other countries.”  Page 2

Suggestion 13: other relevant departments of the priority countries and regions one by one, and focusing on the major economies countries and the cooperate countries and regions with Chinese Belt and Road Initiative

Reply: Thanks for your suggestion and we made the following changes. Change to “other relevant departments of the priority countries and regions one by one, and focusing on the major economies and the countries and regions that cooperate with the Chinese Belt and Road Initiative.”  Page 3

Suggestion 14: so we will focus on H. arator, O. boas and A. majale in this study.

Reply: Thanks for your suggestion. Now, it has been changed to “this study will focus on H. arator, O. boas and A. majale.”  Page 3

Suggestion 15: Thresholds were chosen differently from the H. arator and A. majale, because of the high correlation between the environmental factors of O. boas.

Reply: Thanks for your suggestion, we have made the following modifications. Change to “Thresholds of O. boas were chosen differently from those of H. arator and A. majale, because of the high correlation between the bioclimatic factors.”  Page 4

Suggestion 16: The AUC average values for 10 repetitions of H. arator, O. boas and A. majale were 0.977, 0.959 and 0.972, indicating that it have a high reliable results in predicting the potential distribution

Reply: Thanks for your suggestion. Now, it has been changed to “The average AUC values for 10 repetitions for H. arator, O. boas and A. majale were 0.977, 0.959 and 0.972, respectively, indicating that the model yielded highly reliable results in predicting the potential distribution.”  Page 6

Suggestion 17: Above all, the training results were consistent on the test data. And it showed that, Bio1 and Bio3 were the most important environmental variables in explaining the potential distribution of H. arator, as to O. boas was Bio3, and A. majale were Bio17 and Bio2.

Reply: Thanks for your suggestion. Now, it has been changed to “Overall, the training results were consistent with the test data. They showed that Bio1 and Bio3 were the most important bioclimatic variables in explaining the potential distribution of H. arator;for O. boas, it was Bio3, and for A. majale, the most important were Bio17 and Bio2.”  Page 7

Suggestion 18: And the smallest was H. arator covers about 3.6×105 km2 , accounted for 3.8% and the optimal and moderate suitability mainly in Yunnan and Sichuan (Figure 3-D,E,F and Table 3).

Reply: Thanks for your suggestion. Now, it has been changed to “Moreover, H. arator covers the smallest possible distribution area of approximately 3.6×105 km2, accounting for 3.8%, and the optimal and moderate suitability areas (0.14 ~ 0.39) were mainly in Yunnan and Sichua.”  Page 9

Suggestion 19: Combining the prediction results of Kistner-Thomas (2019) and Aidoo (2022) for P. japonica and O. monoceros and considering the destructive impacts in native or invasive areas of five insects, the spread of them to other parts of the world should be prevented as early.

Reply: Thanks for your suggestion. Now, it has been changed to “Combining the prediction results from this and the previous studies of Kistner-Thomas (2019) and Aidoo (2022) for P. japonica and O. monoceros, the potential suitable areas of the five species were obtained. In addition, considering the destructive impacts in native or invasive areas of these insects, their spread to other parts of the world should be prevented as early as possible.”  Page 10

Suggestion 20: In all potential distribution areas of these species, the medium and high-risk provinces, such as Yunnan, Hunan, Guizhou, should be strengthen the attention and quarantine next, timely monitor and prevent by customs, forestry and other departments.

Reply: Thanks for your suggestion. Now, it has been changed to “In all possible areas, the medium- and high-risk provinces, such as Yunnan, Hunan and Guizhou, should strengthen their quarantine requirements and should focus on enhanced monitoring by customs, forestry and other departments.”  Page 10

Suggestion 21: Now the potential distribution of the five species are relatively large (Fig

ure1, Figure 3-D,E,F), where the environment is similar to their native areas, and more

suitable for them to survival.

Reply: Thanks for your suggestion. Now, it has been changed to “The potential distribution of the five species is relatively large (Figure1, Figure 3-D,E,F); the chances of their survival and establishment are very high in some areas”  Page 11

Suggestion 22: Nevertheless, when we build and continuously complete the database of

quarantine or invasive scarab beetles and do further research, we will constantly turn

from passive to active.

Reply: Thanks for your suggestion. Now, it has been changed to “However, when we build and continuously complete the database of quarantine or invasive scarab beetles and use this information in preventive measures, we will be able to constantly convert the passive information into active action.”  Page 11

Suggestion 23: The existing database of invasive alien species in China and the National Pest Quarantine Information System in China are relatively complete and functional databases, which have the initial strength to improve their international influence [48,49], but we believe that the database of quarantine or invasive alien insects that we build in here may be a powerful tool for us.

Reply: Thanks for your suggestion. Now, it has been changed to “Among the existing databases the “Database of invasive alien species in China” and the “National Pest Quarantine Information System in China” are relatively complete and functional databases, with the initial strength to improve their international influence [54,55]. However, we believe that the database of quarantine or invasive scarab beetles that we have built here and the work that we will conduct with the database may be a powerful tool enabling us to take one more step.”  Page 11

Reviewer 2 Report

1. English writing should be strengthened. 

2. Add a review of relevant research findings in the introduction. 

3. The discussion is inadequate. 

4. Please see the markings for other issues in the PDF file.

Author Response

Cover letter

Thanks to the reviewers for their important suggestions on the effective improvement of this article. The relevant content has been revised, please see the following for details.

This article has been revised by the language editor.

Suggestion 1: Add a review of relevant existing research findings in the introduction.

Reply: Thanks for your suggestion. In this paper, we analyze and discuss the top five species screened from the database of quarantine or invasive scarabe beetles. There are no relevant studies on H. arator, O. boas and A. majale, and the other two species (P. japonica and O. monoceros) we discussed here are mainly use the results of previous studies. The aim was to highlight the potential invasion risks of the top five species in our database in China.

Relevant contents have been added in the preface, including current research (there are few related studies on the three species studied in this paper) and biological information of the species studied and discussed in this paper.

Suggestion 2: The discussion is inadequate.

Reply: Thanks for your suggestion. The key points discussed in this article are further explained, and some other relevant information is added.  Page 10-12

Suggestion 3: The title might be changed to: Using MaxEnt Model to Predict the Potential Distribution of Three Potentially Invasive Scarab Beetle in China. This is an option presented for discussion and is not mandatory.

Reply: Thank you very much. We accept your suggestion. After our discussion, we think your suggestion is more suitable.  Page 1

Suggestion 4: “especially in China” Why?

Reply: Thanks for your suggestion. It serves as an emphasis. According to the previous analysis, China's customs department quarants out a large number of foreign insects every year, and also faces a great challenge of alien invasive insects. Therefore, we mainly focus on the potential distribution in China in this article.  Page 2

Suggestion 5: “the top five species” How is this sorted? Size? Quantity? Importance?

Reply: Thanks for your suggestion. At page 3, we give a brief explanation (“Based on the database, species were selected for analysis, and their potential distribution areas were discussed.”), which is explained in detail in the Materials and Methods section and Table S1.  Page 3

Suggestion 6: “the official websites of the agriculture, forestry and other relevant departments of the priority countries and regions ” Please add citation information such as website.

Reply: Thanks for your suggestion. This website information is more, we put it in Table S1. And other references are not available, because all the information was searched and saved manually one by one.

Suggestion 7: “Top five pests in the database that hadn't invaded in China are P. japonica (61), H. arator (13), O. monoceros (12), O. boas (11) and A. majale (7). Since P. japonica and O. monoceros had already been studied, so we will focus on H. arator, O. boas and A. majale in this study.” You can absolutely analyse these five spe cies, even if they have already been studied. Discuss your results with theirs. This will facilitate the presentation and discus sion of the results.

Reply: Thanks for your suggestion. The experimental scheme related to P. japonica and O. monoceros are relatively complete, and the prediction results are more reliable. Therefore we decided to cite the results of previous studies and conduct the first analysis of the remaining three species.

Suggestion 8: H. arator, O. boas and A. majale.” Full name

Reply: Thanks for your suggestion. It has been revised in the full text.

Suggestion 9: “Environmental variables are widely used to predict the distribution of invasive alien insects at regional and global scales” What you are expressing should be that the survival of invasive insects is closely related to the environment, and the variables mentioned later are often used for related studies.

Reply: Thanks for your suggestion. This sentence has been revised. “The survival of invasive insects is closely related to bioclimatic factors. Bioclimatic variables are widely used to predict the distribution of invasive alien insects at regional and global scales.”  Page 5

Suggestion 10: The case of the first letter should be the same.

Reply: Thanks for your suggestion. It has been revised in the full text.  Page 5

Suggestion 11: “and focus on the distribution in China.” For the study of China, has the model been reconstructed or is it a local amplification of the world results? Please describe.

Reply: Thanks for your suggestion. It was modified to “We used MaxEnt model (version 3.4.1, http://biodiversityinformatics.amnh.org /open_source/maxent/) to predict the potential distribution of H. arator, O. boas and A. majale globally and in China. For the study of China, the model is a local amplification of the world results.”  Page 6

Suggestion 12: “Then we put the results into ArcGIS software (Version 10.7) and reclassified it into four different levels: optimal suitability areas, moderate suitability areas, marginal suitability areas, and unsuitable areas.” Add the citation information  What are the criteria for classification? And add the citation information.

Reply: Thanks for your suggestion. The criteria for classification and citation information has been added. The occurrence probability of four categories has been provided.  Page 6 & 9

Suggestion 13: “The suitable areas of three species tended to spread to the America and Asia.” How was this conclusion reached?

Reply: Thanks for your suggestion. After analysis, this conclusion is not appropriate, so it will be deleted.

Suggestion 14: “And O. monoceros had no potential range in China, but Aidoo (2022) analyzed that, extensive areas of Asia were highly favorable to the pest” and “For instance, Yunnan should focus on P. japonica, H. arator, O. boas and O. monoceros, as the hot and humid environment in Yunnan province is ideal for their survival.” How was this conclusion reached? You have not analysed their environmental adaptability. Furthermore, according to your description, O. monoceros does not have a suitable habitat in China, whic his contradictory if you accept this result.

Reply: Thank you very much, according to your suggestion, we have some thinking. For O. monoceros, Aidoo (2022) analyzed that, extensive areas of Asia were highly favorable to the pest. So we speculate, with the change of climate, there is a possibility of entering Asia, even China. However, the inference that Yunnan should pay more attention to this species is not cautious, so we remove it. Besides, we also added some information about the living environment of alien insects.  Page 11

Suggestion 15: “Above all, there are more species with potential invasion risks in China.” How was this conclusion reached? Compared to which countries? This comparison is not mentioned above.

Reply: Thanks for your suggestion. This conclusion is not based on comparison, but an extrapolation from previous years (Introduction), such as insects accounted for the majority of the species quarantinated by the customs department. This sentence has been changed to "A large number of insects are quarantined at customs ports every year. Thus, in addition to the five species discussed and analyzed in this study, there are more species that may also carry potential invasion risks for China.”  Page 2 & 11

Author Response

Cover letter

Thanks to the reviewers for their important suggestions on the effective improvement of this article. The relevant content has been revised, please see the following for details.

This article has been revised by the language editor.

Suggestion 1: Grammatical errors and strange expressions

Followings are only selected examples.

Simple summary: potential distribution areas of multiple species -→ these species

Abstract: highly diversity group -→ high diverse group, The prediction results shown -→ show Introduction: Scarab beetles (Coleoptera: Scarabaeoidea) is a highly diversity group of insects including a variety of feeding habits, numbers of species and widely distribution [7].

Reply: Thanks for your suggestion. Revisions were made in response to language concerns raised by the reviewers and subsequently revised by the English language editor.

Suggestion 2: Environmental variables in 4 page

Environmental variables used SDM are all climatic variables. Hence, Environmental variables should be changed as climatic variables.

Reply: We accept your suggestion. After our discussion, we think your suggestion is more suitable. Thank you very much.

Suggestion 3: Rephrase in 4 page

Only one variable from each set of the highly cross-correlated variables was retained for further study ,then we selected seven variables (Pearson> 0.8) to be included in the H. arator, five variables (Pearson> 0.95) for O. boas, eight variables (Pearson> 0.8) for A. majale to analysis [28] (Table 1 and Table S1)

Yellow parts may be threshold (Pearson correlation coefficient). These should be commented in previous sentence.

Reply: Thanks for your suggestion. To solve this problem, we added “To establish a high-performance model with fewer variables, the Pearson correlation coefficients of the cross-correlations among the 19 bioclimatic variables were calculated using the SPSS software (Version 20.0), and the Pearson correlation coefficient (threshold) was included in parentheses.”  Page 5

Suggestion 4: Materials and Methods

“Then we put the results into ArcGIS software (Version 10.7) and reclassified it into four different levels: optimal suitability areas, moderate suitability areas, marginal suitability areas, and unsuitable areas.”

The occurrence probability of four categories should be provided.

Reply: Thanks for your suggestion, we have made the following modifications. Because of the 3 species analyzed in this paper differ in occurrence probability, so we annotate them respectively in the Results section.  Page 9

Suggestion 5: Others

5 page: it also appeared to have the maxmum gain reduction when it was omitted.

1-→ it also appeared to have the maxmum gain reduction when it was omitted (Figure 2).

Reply: Thanks for your suggestion. Now it has been revised.

Suggestion 6: Table 2: Remake this table simply. One column one variable.

Reply: Thanks for your suggestion. In order to solve this problem, we made some attempts. Adjust to a column of one variable too long, so each column is still a species, but each row is a variable.  Page 7

Round 2

Reviewer 1 Report

The manuscript flow has improved significantly from the previous version. I just found a few edits below. After this manuscript is good to fly. 

while Oryctes monoceros has no suitable area. 
line 35:  pay more attention to monitoring for the prevention of infestation by invasive insect pests. 
Line 41: Invasive alien species threaten biodiversity around the world. Increasing global trade and 
environmental changes have facilitated the arrival and establishment of invasive species, and as a result, there is a global push toward research on controlling invasive species (1).  
Line 70: reported host plants [10, 12], including 
Line 81:  Both adults and larvae are pests (change this everywhere).  
Line 120: The results obtained in this study will help environmental protection and quarantine agencies to implement quarantine measures that will reduce the possibility of invasion of these insects in China and other countries. In addition, the database introduced in this paper can provide data and information support systems for ensuring national biosecurity.
Line 236: 47.7mm add a unit after numbers. 
Line 265: remove ,"its"
305:  their distribution. 
311: The AUC values for all of them were above 0.9,
remove, "In the updating and 316 improvementation of the database in the future, further data supplement will focus on 317 the new, harmful insects intercepted at customs ports and detected by the agriculture 318 and forestry departments of each province, or the sudden outbreak of harmful scarab 319 beetles abroad with the possibility of global spread." It is not making sense here.
339: remove, taken together.
342: of these five species
352: time to time
functional databases, with implications for pest management around the world (54,55). 
378: more steps towards preventive measures. 

Author Response

Cover letter

Thanks to the reviewers for their important suggestions on the effective improvement of this article. The relevant content has been revised, please see the following for details.

Suggestion 1

while Oryctes monoceros has no suitable area.

Line 35: pay more attention to monitoring for the prevention of infestation by invasive insectpests.

Reply: Thanks for your suggestion. Now these sentences have been revised.  Page 1

Suggestion 2

Line 41: Invasive alien species threaten biodiversity around the world. Increasing global tradeand environmental changes have facilitated the arrival and establishment of invasive species.and as a result there is a global push toward research on controlling invasive species.

Line 70: reported host plants [10, 12], including.

Line 81: Both adults and larvae are pests (change this everywhere).

Reply: Thank you very much, these suggestions are very useful. Now we have finished the revision.  Page 2

Suggestion 3:

Line 120: The results obtained in this study will help environmental protection and quarantineagencies to implement quarantine measures that will reduce the possibility of invasion of theseinsects in China and other countries. in addition. the database introduced in this paper canprovide data and information support systems for ensuring national biosecurity.

Reply: Thanks for your suggestion. This sentence has been revised.  Page 3

Suggestion 4

Line 236: 47.7mm add a unit after number.

Reply: Thanks for your suggestion. Now it has been revised.  Page 7

Suggestion 5

Line 265: remove ."its"

Reply: Thanks for your suggestion. Now it has been revised.  Page 8

Suggestion 6

305: their distribution.

Reply: Thanks for your suggestion. It has been modified to occurrence records.

  Page 10

Suggestion 7

311: The AUC values for all of them were above 0.9 remove,

Reply: Thanks for your suggestion. Now it has been revised.  Page 10

Suggestion 8

"In the updating and 316 improvementation of the database in the future, further datasupplement will focus on 317 the new. harmful insects intercepted at customs ports and detectedby the agriculture 318 and forestry departments of each province. or the sudden outbreak ofharmful scarab 319 beetles abroad with the possibility of global spread." it is not making sensehere.

Reply: Thank you very much. We accept your suggestion. After our discussion, we think your suggestion is more suitable, and we put it in a new position.  Page 11

Suggestion 9

339:remove taken together

342: of these five species

352: time to time

functional databases, with implications for pest management around the world (54.55)

378: more steps towards preventive measures.

Reply: Thanks for your suggestion. These questions have been revised.  Page 11

Author Response

Cover letter

Thanks to the reviewers for their important suggestions on the effective improvement of this article. The relevant content has been revised, please see the following for details.

Suggestion 1: the occurrence probability for four categories of potential distribution should be provided in materials and methods or in figure caption (this was suggested in first review)

Reply: Thanks for your suggestion. I am sorry that the previous modification did not understand this suggestion well. So this revision has been deeply considered on it.

As for the occurrence probability , we have the following views:

(1). The occurrence probability means the range of the suitablility. Because we analyzed three species in this paper, the classification standard are not consistent, so the occurrence probability is not consistent. It has been supplemented in the notes of Figure 3. The details are as follows.  Page 9

“* (A,D): For Heteronychus arator, the range of the suitablility (occurrence probability) from unsuitable to optimal suitability is 0 ~ 0.05, 0.05 ~ 0.18, 0.18 ~ 0.40, 0.40 ~ 0.82 in the world and 0 ~ 0.04, 0.04 ~ 0.14, 0.14 ~ 0.25, 0.25 ~ 0.39 in China; (B,E): For Oryctes boas, from unsuitable to optimal suitability is 0 ~ 0.08, 0.08 ~ 0.26, 0.26 ~ 0.49, 0.49 ~ 0.88 in the world and 0 ~ 0.02, 0.02 ~0.10, 0.10 ~ 0.19, 0.19 ~ 0.53 in China; (C,F): For Amphimallon majale, from unsuitable to optimal suitability is 0 ~ 0.06, 0.06 ~ 0.21, 0.21 ~ 0.43, 0.43 ~ 0.85 in the world and 0~ 0.03, 0.03 ~ 0.11, 0.11 ~ 0.20, 0.20 ~ 0.38 in China.”

(2). Classification standard. We used the" Natural Breaks [Jenks]" method to reclassify the suitable ranges of the three species in the world and in China.   Page 6 - Line 202 -205)

Suggestion 2:

L13: were → was;

L51-52: “Scarab beetles (Coleoptera: Scarabaeoidea) are a highly diverse group of insects with a number of feeding habits and species and a wide distribution”

Yellow part may not be needed;

L137: to obtain a the preliminary -→ to obtain a preliminary

Reply: Thanks for your suggestion. Now it has been revised.  Page 1, 2, 4

Suggestion 3: L140: have already been studied.

References should be needed after this sentence

Reply: Thanks for your suggestion. The references had been added.

Suggestion 4: L264-287: optimal suitability (0.49 ~ 0.88), optimal suitability (0.43 ~ 0.85), moderate suitability areas (0.11 ~ 0.38), optimal and moderate suitability areas (0.10 ~ 0.53) optimal and moderate suitability areas (0.14 ~ 0.39)

What mean the different values in parentheses?

Reply: Thanks for your suggestion. the differernt values means different range of suitability value. Now these have been removed, because the notes of Figure 3 list the complete content of this section.

Suggestion 5: L291-292: “Figure 3. Potential distribution of Heteronychus arator, Oryctes boas and Amphimallon majale in the world (A,B,C) and in China (D,E,F).”

Provide the occurrence probability four categories in this figure caption.

Reply: Thanks for your suggestion. We used the" Natural Breaks [Jenks]" method to reclassify the suitable ranges of the three species (it also can see at Page 6 - Line 202 -205). And the occurrence probability four categories has been added in Figure 3.  Page 9
